# Advances in Poultry Vaccines: Leveraging Biotechnology for Improving Vaccine Development, Stability, and Delivery

**DOI:** 10.3390/vaccines12020134

**Published:** 2024-01-28

**Authors:** Khaled Abdelaziz, Yosra A. Helmy, Alexander Yitbarek, Douglas C. Hodgins, Tamer A. Sharafeldin, Mohamed S. H. Selim

**Affiliations:** 1Department of Animal and Veterinary Science, College of Agriculture, Forestry and Life Sciences, Clemson University Poole Agricultural Center, Jersey Ln #129, Clemson, SC 29634, USA; 2Clemson University School of Health Research (CUSHR), Clemson, SC 29634, USA; 3Department of Veterinary Science, Martin-Gatton College of Agriculture, Food, and Environment, University of Kentucky, Lexington, KY 40546, USA; yosra.helmy@uky.edu; 4Department of Animal & Food Sciences, University of Delaware, 531 S College Ave, Newark, DE 19716, USA; yitbarek@udel.edu; 5Department of Pathobiology, Ontario Veterinary College, University of Guelph, Guelph, ON N1G 2W1, Canada; dhodgins@uoguelph.ca; 6Department of Veterinary Biomedical Science, Animal Disease Research and Diagnostic Laboratory, South Dakota State University, Brookings, SD 57007, USA; tamer.sharafeldin@sdstate.edu (T.A.S.); mohamed.selim@sdstate.edu (M.S.H.S.)

**Keywords:** poultry, chicken, biotechnology, vaccine, inactivated, attenuated, DNA, mRNA, subunit, recombinant, adjuvant, stability, delivery, in ovo, nanoparticles

## Abstract

With the rapidly increasing demand for poultry products and the current challenges facing the poultry industry, the application of biotechnology to enhance poultry production has gained growing significance. Biotechnology encompasses all forms of technology that can be harnessed to improve poultry health and production efficiency. Notably, biotechnology-based approaches have fueled rapid advances in biological research, including (a) genetic manipulation in poultry breeding to improve the growth and egg production traits and disease resistance, (b) rapid identification of infectious agents using DNA-based approaches, (c) inclusion of natural and synthetic feed additives to poultry diets to enhance their nutritional value and maximize feed utilization by birds, and (d) production of biological products such as vaccines and various types of immunostimulants to increase the defensive activity of the immune system against pathogenic infection. Indeed, managing both existing and newly emerging infectious diseases presents a challenge for poultry production. However, recent strides in vaccine technology are demonstrating significant promise for disease prevention and control. This review focuses on the evolving applications of biotechnology aimed at enhancing vaccine immunogenicity, efficacy, stability, and delivery.

## 1. Introduction

The poultry industry is the largest contributor of meat for human consumption worldwide. Table 1 provides a list of various avian species commonly involved in the poultry industry.

As the poultry industry rapidly expands, the increase in high-density poultry farms raises the risk of disease outbreaks [1]. Over the last few decades, the poultry industry has witnessed significant challenges. Foremost among these challenges are preventing and controlling poultry diseases that impact animal welfare and cause significant production losses and improving consumer confidence in the safety and quality of poultry products by mitigating the risk of food-borne pathogens. Examples of infectious diseases that form a continuous threat to the poultry industry include Newcastle disease (ND), avian influenza (AI), infectious bursal disease (IBD), arthritis/tenosynovitis caused by reovirus, infectious bronchitis (IB), infectious laryngotracheitis (ILT), Marek’s disease (MD), fowl pox (FP), chicken anemia, Mycoplasmosis caused by *Mycoplasma gallisepticum*, necrotic enteritis (NE) caused by *Clostridium perfringens,* and coccidiosis caused by *Eimeria* spp. [2,3,4,5,6,7].

Despite strong efforts toward applying biosecurity practices in developed poultry industries, in 2016, according to the Food and Agriculture Organization (FAO) [8], the cost triggered by poultry diseases was estimated to be 20 percent of the gross value of poultry production and was expected to be higher in developing countries. Furthermore, the costs of antibiotic-resistant infections also constitute a significant economic burden [9]. Therefore, establishing integrated, cost-effective disease control systems by developing prophylactic and therapeutic vaccines is crucial to preventing or decreasing the potential for disease emergence at the farm level, thus reducing the economic losses incurred by disease outbreaks.

Until recently, vaccine technologies encountered many technical challenges associated with vaccine formulation, delivery, and the durability of protection. Historically, the conventional vaccine platform was based on attenuation or inactivation of the pathogen, often referred to as first-generation vaccines, so that it elicits protective immunity without causing disease [1,10]. Despite their traditional manufacturing processes, conventional vaccines are still extensively used in the poultry industry to combat several viral diseases such as MD, ND, IB, IBD, ILT, and FP, bacterial diseases such as fowl cholera and salmonellosis, and parasitic diseases such as coccidiosis. However, the poor immunogenicity of inactivated vaccines and their limited ability to stimulate T cell-mediated components of the adaptive immune response, especially those with high antigenic diversity, as well as the instability of live attenuated vaccines, may pose a problem in achieving optimal vaccine impact. Some of these shortcomings were later addressed by adding natural or synthetic adjuvants to the inactivated vaccines to enhance their immunogenicity and by lyophilizing live attenuated vaccines to achieve adequate storage stability. However, despite these achievements, some conventional vaccines necessitate booster doses to sustain long-lasting protective immunity [1].

Efforts are being made to eliminate the need for booster doses by formulating vaccines and adjuvants in biodegradable and biocompatible natural and synthetic polymer-based particulate vaccine carriers. In addition to protecting the encapsulated vaccine, these microparticles/nanoparticles are engineered to facilitate a gradual release of vaccine antigens, thereby providing sustained immune responses [11,12]. Various micro/nanoparticles are being investigated as delivery systems, including poly (D, L-lactide-co-glycolide) (PLGA), alginates, chitosan, liposomes, and, most recently, polyphosphazenes [11,13,14,15,16].

The rapid pace of technologies has played a crucial role in tackling most poultry industry research challenges and driving tangible improvements in vaccine research and development. For instance, recent advances in molecular techniques such as recombinant DNA technology, polymerase chain reaction (PCR), and high-throughput genome sequencing platforms have revolutionized approaches to vaccine design and development as well as the evaluation of vaccine efficacy, mechanisms of action, and vaccine-mediated immune activation. This has resulted in the discovery of new-generation vaccines, which rely on specific pathogen-derived molecules or utilize recombinant DNA and RNA technologies [1].

This review presents various novel approaches that have incorporated the use of new technologies for improving vaccine immunogenicity and efficacy, as well as the creation of potent adjuvants. Furthermore, the role of biotechnology in employing advanced and scalable methods for vaccine stabilization and delivery will be discussed.

## 2. Vaccine Formulations

The evolution of vaccines can be traced back to the observation in the 18th century that a biological preparation of weakened or killed forms of the microbe can confer a protective immunity against a specific disease [17]. Over time, vaccine technology has yielded a wide range of new vaccines. Currently, vaccines can be classified, according to the nature of the antigen, into three groups: (a) first-generation (conventional) vaccines, which include live attenuated and inactivated or killed vaccines; (b) second-generation vaccines, also called subunit vaccines, which are composed of mostly protein parts such as protein antigens or recombinant proteins, and (c) third-generation vaccines which include recombinant vector and nucleic acid (DNA and mRNA) vaccines. Appendix A provides a list of current commercial poultry vaccines, including their trade names, manufacturers, and routes of administration, in accordance with the latest updates from the Canadian Food Inspection Agency.

### 2.1. First-Generation (Conventional) Vaccines

#### 2.1.1. Live Attenuated Vaccines

Attenuation is one of the most used approaches for producing poultry vaccines. Classically, the manufacture of live attenuated vaccines required the use of laboratory animals or chicken embryos to attenuate the virulence of a pathogen. The first live attenuated poultry vaccine was developed by Louis Pasteur in the 1880s against fowl cholera by the repeated passage of the bacterium *Pasteurella multocida* in laboratory animals to reduce its virulence [17]. The establishment of cell culture technology in the 1950s opened new avenues for using animal cells to attenuate the pathogen as an in vitro model alternative to the animal [18,19]. Following this breakthrough, the Animal Welfare Act in 1966 (Laboratory Animal Welfare Act, P.L. 89-544) [20] mandated researchers to identify alternatives to animal experimentation. Thanks to technology, researchers now have alternative options to the experimental procedures that cause distress to animals. Since then, cell culture-based technology has been widely used to produce various poultry vaccines. An example of a commercially available cell-culture-derived, live attenuated vaccine is the ILT vaccine, a modified live virus [21].

Despite the undeniable success of live attenuated vaccines in generating long-lasting immunity (both cell- and antibody-mediated immunity) and controlling many infectious diseases in poultry, the likelihood of reversion to virulence remains among the top concerns associated with these vaccines. A typical example is the recurrent ND, AI, and IBD outbreaks in vaccinated chickens [1].

The advent of genetic mapping and the establishment of reverse genetics in 1995 marked a turning point in live vaccine technology [22]. The reverse genetics-based strategy has been utilized to generate genetically modified live attenuated vaccines as safer alternatives to live vaccines. This process involves the manipulation of the viral genome either by genetic re-assortment, which leads to the production of new variants with distinct gene constellations, or by the introduction of mutations into single or multiple internal genes, leading to the generation of a virulent virus with impaired ability to replicate in the host while retaining its immunogenicity [23]. Despite the promising evidence that genetically modified live vaccines are effective against various viral diseases, such as ND, AI, IBD, ILT, and MD [1,24,25,26], their commercial availability remains murky.

#### 2.1.2. Inactivated Vaccines

The development of inactivated-whole cell vaccines started in the late 18th century by killing the pathogen using physical (heat) or chemical (formaldehyde, diethylpyrocarbonate, and β-propriolactone) processes to denature the proteins or damage the nucleic acids, thus eliminating its infectivity [1,27,28]. However, some disadvantages are still associated with these inactivation methods, including their impact on the antigenic structure of the pathogen and the more significant variability in reproducibility between inactivation methods, in addition to the potential contamination with chemical residues.

Technology has played a role in the production of inactivated vaccines. Recently, irradiation technology, such as gamma-irradiation, has emerged as a fast and safe alternative to inactivate organisms with limited or no effects on antigenic determinants [29]. Despite these advances, no commercial irradiated vaccines are currently available for commercial poultry farming. Research is underway to examine the potential of several promising irradiated vaccines against various pathogens, including viral, such as AIV [30]; bacterial, such as *Salmonella enterica* var. Typhimurium [31]; and protozoal, such as *Eimeria* spp. [32,33].

In spite of the widespread use of conventional vaccines, the recurrent alterations in the antigenicity of certain pathogens due to antigen drift and shift pose a potential risk of future vaccination failures with these vaccines. Therefore, constant vaccine evaluation and updates are required to accommodate the frequent antigenic changes. Indeed, modern biotechnology and genomic tools have provided a means for generating new generation vaccines to overcome the drawbacks associated with conventional vaccines. Figure 1 presents a schematic diagram depicting various categories of first-generation vaccines.

### 2.2. Second-Generation Vaccines

#### 2.2.1. Conventional Subunit Vaccines

Unlike conventional whole-cell vaccines, non-replicating subunit vaccines contain only the pathogen’s antigenic parts (epitopes) as immunogens. The production of traditional subunit vaccines involves the propagation of the pathogen and subsequent extraction of the various potential antigenic determinants that can effectively elicit a potent immune response, such as viral glycoproteins and bacterial whole-cell proteins, outer membrane proteins (OMPs), purified flagellar proteins, fimbrial proteins, pilus proteins, and lipopolysaccharides (LPS) [1,10]. Recombinant subunit vaccines have also been developed against various bacterial, viral, and parasitic pathogens by cloning the gene that encodes specific antigens, followed by their expression in an appropriate system, such as bacterial, yeast, insect cells, or mammalian cells [34,35,36]. Subsequent steps involve isolation, purification, and formulation of the expressed proteins. Despite scientific consensus that these vaccines eliminate the risks associated with live vaccines, the essential need for adjuvants to enhance their immunogenicity coupled with the costs associated with the massive production of the antigenic proteins makes their application economically infeasible.

#### 2.2.2. Recombinant Subunit Vaccines

The development of recombinant DNA technology in the 1970s has provided a novel tool for the large-scale production of protein antigens [37,38,39]. The production of recombinant proteins is typically achieved by nucleotide sequencing of the genome of the infectious agent, followed by identification of the gene(s) encoding the protein antigen and then insertion of the candidate gene into mammalian, yeast, or bacterial cells for the expression of the target protein. The expressed proteins are further extracted and purified to produce a subunit vaccine. Considerable research has been devoted to evaluating the efficacy of recombinant subunit vaccines against multiple pathogens, such as IBDV [40], *Eimeria* species [41,42], chicken infectious anemia [43], and avian leukosis virus (ALV) [44].

Even though this technology enables high protein yields and allows the incorporation of multiple proteins, rich in defined epitopes, into a subunit vaccine, the lack of exogenous immune activating components, such as lipids, nucleic acid, etc., makes them moderately immunogenic [45]. This problem was partially overcome by incorporating adjuvants into these vaccines to enhance their immunogenicity. For example, a quadrivalent subunit vaccine containing recombinant proteins for four *Eimeria* species (*E. tenella*, *E. necatrix*, *E. acervuline*, and *E. maxima*) provided partial cross-protection against heterologous challenge with these species [45]. On the other hand, an enhanced protective immunity against avian coccidiosis [41,42], chicken anemia virus [42], and ALV [44] was attained when recombinant antigens were used in combination with different adjuvants. Nonetheless, the possible occurrence of improper protein folding and inclusion body formation during protein translation constitutes a barrier to the successful development of recombinant vaccines. Figure 2 presents a schematic diagram depicting various categories of second-generation vaccines.

### 2.3. Third-Generation Vaccines

Third-generation vaccines employ DNA (as plasmids) and RNA (as mRNA) to generate antigens of interest by the host cell machinery, followed by activation of the immune response and establishment of memory. In addition to their vastly easier and rapid production, there are various advantages of third-generation vaccines over conventional vaccines (live-attenuated vaccines, inactivated vaccines, split-virion influenza vaccines, and subunit vaccines): (a) Third-generation vaccines are safer than the live attenuated vaccine. For instance, the use of live attenuated vaccines against highly pathogenic viruses is normally hampered by the potential of reversion to virulent pathogens causing illnesses in the host. (b) In the case of inactivated vaccines and subunit and recombinant protein-based vaccines, adjuvants are required to enhance immunogenicity, which is normally not required for third-generation vaccines, especially mRNA vaccines after significant modifications were made to the mRNA chemistry that modulates mRNA stability, innate immune activation and the level of translation [46]. (c) Third-generation vaccine production does not necessitate higher biosafety level laboratories, as needed for conventional vaccines. (d) Additionally, as was observed with the mRNA vaccines used against Severe Acute Respiratory Syndrome Coronavirus 2 (SARS-CoV-2), there has been a faster registration and approval time for third-generation vaccines compared to conventional vaccines, with approximately 10 years required for conventional vaccines and only 18 months for mRNA vaccines [47]. (e) Lastly, vaccine constructs used in third-generation vaccines can only include the key antigen of interest. This approach prevents any adverse impact from potentially harmful proteins, like toxins or non-essential immunodominant proteins that do not contribute to protection [48]. These advantages make third-generation nucleic acid vaccines promising and versatile vaccines to control infectious disease outbreaks in humans and animals. Below, recombinant vector, DNA, and RNA (specifically mRNA) vaccines will be reviewed for their potential in controlling poultry pathogens, including those that have a zoonotic potential for human infections.

#### 2.3.1. Recombinant Vector Vaccines

Advances in molecular biology have made it possible to sequence the genomes of viral pathogens rapidly and economically. Genes that code for critical virulence factors can be identified, as well as those genes that can be deleted from a genome without affecting viral replication. Genes that are not essential for replication (in cell culture and/or in vivo) can be deleted and replaced by specific gene segments coding for proteins (virulence factors) from unrelated pathogens. Viruses, which can be engineered to express virulence factors from other pathogens, act as viral vectors [49]. Viruses with relatively large double-stranded DNA genomes, such as pox viruses, herpes viruses, and adenoviruses, are widely used as viral vectors. In some cases, attenuated live vaccines that have been used successfully for years (e.g., herpesvirus of turkeys (HVT)) can be genetically modified to induce protective immune responses to additional pathogens, thus generating bivalent or trivalent vaccines. The discovery of optimal site (s) for the insertion of a foreign gene, the selection of appropriate gene segments to insert into a viral vector, and the choice of promoter sequences to drive expression are critical to the success of these vaccines [50,51]. Commonly used viral vectors have been reviewed in detail by Romanutti et al. [52].

The ability to engineer bivalent and trivalent vaccines opens the possibility of simpler vaccination protocols. Commercial trivalent vaccines have been licensed for vaccination of chicks (in ovo or at 1 day of age) that consist of an HVT vector expressing antigen of IBDV with antigen of either NDV or LTV. The viral vector itself has immune-stimulating (adjuvant) effects and promotes both antibody- and cell-mediated immune responses. For some viruses, cell-mediated responses are critical for protection; killed vaccines rarely induce cell-mediated immunity.

Viral vector vaccines should have superior safety profiles because inserting a gene that codes for a single protein and not for an entire virus into a viral vector eliminates the inherent risk of reversion to virulence associated with modified live vaccines. For some pathogens, it is important to be able to distinguish serological responses due to vaccination from responses due to exposure and infection with the specific pathogen. Vaccination against highly pathogenic avian influenza (HPAI) is of limited practical value if, following vaccination, it is no longer possible to monitor serologically for infection with the virulent circulating strain [53]. Viral-vectored vaccines expressing a single protein of HPAI can be used to induce both antibody- and cell-mediated immune responses. Because vectored vaccines do not induce antibodies to diverse other components of the virus, these vaccines are DIVA (differentiating infected from vaccinated animals) and are useful during disease outbreaks or for epidemiological studies to design disease eradication measures.

Antibodies produced by a breeder hen in response to vaccination are transferred to the yolk during passage down the oviduct. These maternally derived antibodies (MDA) are critical for protecting chicks in the first days after hatch. The concentration of these antibodies in the serum of the chicks varies depending on vaccination programs but declines steadily until it is no longer protective. Unfortunately, MDA can inhibit the replication of modified live virus vaccines and can limit antibody responses to killed vaccines. Bertran et al. compared the effects of MDA specific for an NDV vector and MDA specific for the H5 antigen of AIV on H5-specific antibody responses of broiler chicks vaccinated at 1 day of age and evaluated protection against viral challenge [54]. MDA to the vector and MDA to the H5 insert both suppressed antibody responses; vaccination failed to protect against the challenge virus. In contrast, vaccination with an HVT vector expressing H5 antigen-induced antibody responses and protection against challenge. Romanutti and colleagues have suggested that HVT and other MD virus vectors are resistant to the effects of MDA because the vector virus is cell-associated [52].

#### 2.3.2. Nucleic Acid Vaccines

Nucleic acid vaccines differ from those using recombinant vectors, as they consist only of DNA or RNA. In addition to infectious diseases, nucleic acids have been studied in the context of cancer research, where DNA and mRNA vaccines encoding a broad range of tumor antigens are used. Extensive reviews of nucleic acid vaccines have been conducted previously [55,56]. While the principles of nucleic acid vaccines for cancers and infectious diseases are similar, this review mainly focuses on infectious diseases in poultry.

##### DNA Vaccines

DNA vaccine development employs the replicative capacity of bacterial-derived plasmids and the insertion of a gene encoding the antigen of interest under the control of a promoter, usually the CMV promoter [57,58]. Historically, the concept of DNA administration to illicit protein expression was demonstrated in the early 1990s with the administration of DNA molecules for the expression of chloramphenicol acetyltransferase, luciferase, and galactosidase in mouse skeletal muscle, resulting in efficient protein expression [59]. Following this, a DNA vaccine against viral nucleoprotein of the influenza virus was demonstrated to have a protective effect against challenge with a heterologous strain of influenza A virus in mice, with lower viral titer in the lung, reduced mass loss, and increased survival observed in mice challenged with the virus [60]. Soon after, DNA vaccine efficacy was demonstrated in avian species.

In 1999, immunization of chickens with plasmid DNA encoding infectious IBD virus antigen showed a protective efficacy against infection with this virus in chickens [61]. Hemagglutinin DNA vaccine from A/Turkey/Ireland/83 (H5N8) conferred complete immune protection against the same virus and 95% cross-protection against other two H5N2 antigenic variants with 11% to 13% amino acid variability in the antigenic region (A/Chick/Pennsylvania/1370/83 (H5N2) and A/Chick/Queretaro/19/95 (H5N2), respectively) [62]. A study by Li et al. showed that intramuscularly inoculated quails with 10, 15, 30, or 60 µg of plasmid expressing an HA gene of the H5N1 virus A/goose/Guangdong/1/96 (GS/GD/96) strain protected the birds following challenge with a homologous virus three weeks post-vaccination [63]. The same study showed that while birds not vaccinated had 100% mortality, and those vaccinated with 10 µg of plasmid DNA had a mortality of 20%, all other doses resulted in 100% livability, which was also accompanied by 100% seroconversion in those receiving 30 and 60 µg plasmid and 60–70% seroconversion in those receiving 10 and 15 µg plasmid DNA. In other avian diseases, DNA vaccines have been shown to be efficacious against the ILT virus [64,65], *Eimeria* spp. [66], IB virus [67,68], and ND virus [69]. In other avian species, Triyatni et al. demonstrated the effectiveness of a DNA vaccine that utilized the large (pre-S/S) and small (S) surface proteins of duck hepatitis B (DHB) virus, resulting in the development of total anti-DHB and specific anti-S antibodies and clearance of the virus following infection [70].

While the mechanisms of action of DNA vaccines are not fully understood, current evidence suggests more than one mechanism of action working synergistically. The mechanisms include the expression of DNA by somatic cells such as myocytes and presentation to CD8+ T cells via the MHC I complexes, antigen presentation by professional antigen-presenting cells (APCs) such as dendritic cells (DCs) following transfection by the administered vaccine DNA, and when APCs phagocytize vaccine DNA transfected somatic cells resulting in cross-priming and presentation of antigens to both CD4+ and CD8+ T cells [71].

Different DNA vaccines have been successfully licensed for use against veterinary pathogens. For example, DNA vaccines for West Nile virus (WNV) prevention in horses [72], a fish vaccine for infectious hematopoietic necrosis virus in schooled salmons [73], a dog cancer immunotherapeutic vaccine against melanoma [74], and a plasmid that encodes growth hormone-releasing hormone (GHRH) given via electroporation to pregnant sows to prevent fetal loss [75]. However, it is worth noting that no DNA vaccines have been authorized for use in poultry.

##### RNA Vaccines

Messenger RNA (mRNA) vaccines, which direct the synthesis of protein antigens, have successfully been used against the current human pandemic SARS-CoV-2, a strain of coronavirus that causes coronavirus diseases 2019 (COVID-19). In 1993, Martinon et al. showed that influenza virus-specific cytotoxic T lymphocytes were generated in vivo in mice vaccinated with liposome-entrapped mRNA encoding viral nucleoprotein, although protection was post-challenge with influenza virus was not tested [76]. While the potential of mRNA as a vaccine has been researched for more than three decades, with the first attempt made in the 1990s [59], the vaccine did not succeed until a critical modification was made that increased its stability and translational capacity [77]. The critical modifications included the use of modified nucleosides, which results in reduced activation of dendritic cells, thereby reducing the immunogenicity of the mRNA. In addition, the use of pseudouridine in place of uridine has also resulted in better mRNA stability and translation capacity [77,78].

Compared to conventional vaccines, mRNA vaccines are advantageous because of their high potency, potential for low manufacturing cost, rapid development and scalability, and safe administration [79]. Furthermore, a competitive advantage of mRNA vaccines includes the similarity in the production and purification processes regardless of the encoded antigen, providing a unified protocol for different mRNA vaccine construct production as soon as the genomic sequences of the target antigens are known [80,81]. Compared to DNA vaccines that need to be delivered into the nucleus via electroporation or other mechanisms, mRNA vaccines can be delivered via a regular needle injection as they only need to be delivered to the cytoplasm for translation, a feature that has resulted in increased interest in mRNA vaccines.

Two categories of mRNA vaccine constructs are being actively evaluated, namely the non-replicating mRNA (NRM) and the self-amplifying mRNA (SAM) constructs [79]. While both NRM and SAM contain structures that mimic endogenous mRNA, such as a 5′cap, 5′ and 3′ untranslated regions, an open reading frame encoding the antigen, and a 3′ poly(A) tail, SAM constructs also contain an additional structure: a helicase that encodes for an RNA-dependent RNA polymerase. The helicase structure allows SAM’s self-amplification, resulting in the generation of large amounts of antigens with a low mRNA vaccine dose [82]. However, the NRM seems to have progressed the furthest into clinical practice and application so far.

Due to the delicate nature of the RNA, delivery systems have been devised to safeguard and stabilize the mRNA constructs from degradation during applications and enhance their bioavailability, thereby resulting in a robust immune response [83,84]. Recent breakthroughs in cutting-edge nanotechnology have revolutionized vaccine delivery platforms. For instance, nanoparticle (NP)-based technologies have emerged as promising replacements for older vaccine delivery methods [85,86]. Substantial progress has been achieved in the development and application of NP delivery technologies, including lipid-based nanoparticles (LNPs), carbon nanotubes, polyplexes, polymeric nanoparticles, hydrogel beads, and colloidal nanoparticles made from Generally Recognized As Safe (GRAS) polysaccharides and proteins (e.g., alginate, chitosan, and gelatin) and other delivery methods, such as squalene-based cationic nano-emulsions [79,87,88,89]. However, phospholipids, cholesterol, and polyethylene glycol (PEG)-containing LNPs are the most commonly used delivery systems, while the others are still in their developmental and optimization stages [79].

The major mechanism of action of mRNA vaccines is through the translation and processing of antigenic proteins in the cytoplasm and the presentation of antigens via the MHC class I and II. Once the administered mRNA molecule is released from the delivery system (such as the LNP) in the cytoplasm, it has three major functions and pathways. One pathway is the induction of type I interferons via the activation of pathogen recognition receptors (PRRs), such as Toll-like receptors (TLRs) and other cytosolic PRRs, which creates a T helper (Th)1 favored response. Another pathway the mRNA follows is the processing of the translated protein into polypeptides by the proteasome, which is then presented on the MHC-I complexes on the cell surface. Finally, another mechanism is the secretion of a folded protein that is then absorbed by APCs, processed, and presented on MHC-II complexes [80]. The mechanisms of action of mRNA vaccines are depicted in Figure 3.

Recent studies have shown the efficacy of mRNA vaccines against viral and bacterial pathogens. A study by Arevalo et al. showed that intramuscular immunization of mice with a cocktail of multivalent nucleoside-modified mRNA vaccine encoding a single full-length HA of all 20 known influenza virus subtypes induced high levels of antibodies against individual HA [90]. The same study showed that the cocktail of mRNA-LNP was not inducing antibodies that were broadly cross-protective but instead elicited strain-specific responses against each strain. While studies in poultry are limited, recent findings in chickens show promising outcomes using mRNA vaccines against viral pathogens. A recent study by Xu et al. demonstrated the efficacy of an HA mRNA transcript from H9N2 and encapsulated in LNP [91]. The study demonstrated that the vaccine was safe in embryonated and live chicken and that the mRNA vaccine at 25 μg dose showed comparable antibody titers to an inactivated vaccine, while titers for 10 and 15 μg mRNA were significantly lower than 25 μg and the inactivated vaccine. ELISPOT assay showed that, while all mRNA vaccine doses had significantly higher IFN-positive cell counts compared to the inactivated vaccine, a dose-dependent response was observed in the mRNA vaccine with a 25 μg vaccine dose showing 2× the IFN-γ positive cell compared to a 15 μg vaccine dose [91]. Another study in organoids and embryonated chicken eggs has demonstrated the effective translation of mRNA vaccines [91]. In addition to viral pathogens, where most of the research with mRNA vaccines has focused, a recent study by Kon et al. demonstrated that a nucleoside-modified mRNA-LNP vaccine based on the bacterial F1 capsule antigen, a major protective component of *Yersinia pestis*, elicited humoral and cellular immunological responses in C57BL/6 mice and conferred rapid, full protection against lethal *Y. pestis* infection after a single dose [92]. This research highlighted the potential role of mRNA vaccines in controlling bacterial pathogens, which can be translated to poultry pathogens that have been negatively impacting the sustainability of poultry production, such as necrotic enteritis. Future research should explore the role of mRNA vaccines to control both viral and bacterial pathogens that have been negatively impacting the poultry industry. Figure 4 presents a schematic diagram depicting various categories of third-generation vaccines.

## 3. Vaccine Adjuvants

The multiple downsides of traditional vaccines (either live attenuated or killed vaccines) triggered the researchers to develop a new line of vaccines, such as subunit vaccines (toxoid and virus-like particles). This new line of vaccines could overcome all the disadvantages of traditional vaccines. However, the main issue of this vaccine is its purified nature and lack of ability to replicate, which makes it less immunogenic. Therefore, to avert these problems, adjuvants should be co-administered with the subunit vaccines to boost their immunogenicity [93].

Adjuvants are substances that are added to the vaccine formulations to enhance the immunogenicity and stability of antigens, allowing for a reduction in the required vaccine dosage and production costs [94]. Generally, they function through immunomodulation by triggering the expression of cytokines and chemokines and recruitment of immune cells, improving antigen uptake and presentation, facilitating antigen transport to lymphoid organs, and activating the differentiation of cytotoxic T-cells. Some adjuvants also create a depot of antigen at the injection site, ensuring a prolonged release of antigen, which in turn results in sustained immune responses. The detailed mechanisms of both commercially available and experimentally studied adjuvants have been reviewed elsewhere [95,96].

Given that the immunogenicity of antigens in inactivated and subunit vaccines is generally lower than that in live and modified live vaccines, adjuvants are primarily combined with these vaccines to elicit a more potent immune response [96,97]. However, it should be noted that many of the adjuvants employed in the vaccine formulations for meat-type birds may have adverse effects on the humans consuming the meat of vaccinated birds [98]. Therefore, adjuvants utilized in poultry vaccines, particularly those administered to broiler chickens, should possess a distinctive capability to be absorbed from the injection site with minimal residue in the meat while enhancing the immunoactivity of the vaccines [99].

Since the discovery of adjuvants in 1920, numerous materials, spanning from organic and inorganic to synthetic and natural substances, have been thoroughly researched and shown the capacity to act as potent adjuvants. Adjuvants can be categorized into two types based on their mechanisms of action: particulate (delivery systems) and non-particulate (immune potentiators/immunostimulatory) [95]. Immunostimulatory adjuvants encompass substances like saponins, TLR agonists, and cytokines, while delivery adjuvants include emulsions, microparticles, and mineral salts [95]. Immunostimulatory adjuvants function mainly by promoting the APCs and increasing the secretion of cytokines and other effector molecules, while the main function of the delivery adjuvants is to preserve the conformation of the antigens for proper presentation to the APCs in addition to creating depots at the site of injection to prolong the duration of immune system stimulation [95,100]. Figure 5 presents a schematic diagram depicting various categories of adjuvants.

### 3.1. Conventional Adjuvants

#### 3.1.1. Mineral Salts

Aluminum salt and, to a lesser extent, calcium salts are being used as adjuvants in poultry vaccines [101]. Their primary function revolves around delivering and preserving the vaccine antigens while facilitating depot formation at the injection site. This, in turn, leads to the elicitation of a prolonged immune response. Furthermore, they are considered safe and cost-effective and have the capability to stimulate humoral immune responses. While aluminum salts offer numerous advantages [95], drawbacks are associated with their use as adjuvants. These include their limited capacity to stimulate cellular immune responses and the potential for inflammation and granuloma formation at the injection site, potentially resulting in a deterioration of meat quality [102,103].

#### 3.1.2. Emulsions

Emulsion adjuvants primarily form when two immiscible liquids are combined, with one liquid creating small droplets dispersed within the other and stabilized by a surfactant layer. Indeed, emulsion adjuvants exhibit greater potency than aluminum salts, as they can improve vaccine-induced immunity (cellular and humoral) while inducing long-lasting immunity, making them a suitable choice for animal vaccines [104]. There are three types of emulsion adjuvants: water-in-oil emulsions, oil-in-water emulsions, and water-in-oil-in-water emulsions.

Water-in-oil (W/O) emulsion is formed by dispersing water droplets within a continuous oil phase. The antigen is entrapped in the water droplets surrounded by a continuous oil phase, resulting in the slow release of antigens upon oil breakdown after injection. The (W/O) emulsion is being commercialized under the product name Montanide™ Incomplete SEPPIC Adjuvants (ISA) (SEPPIC, France) [105]. Montanide™ is a unique emulsions adjuvant that has been used for the production of several poultry vaccines to provoke both humoral and cellular immunity together with long-term protection against several diseases such as ND and AI [106,107,108,109] and Avian borreliosis [110]. Furthermore, Montanide™ (ISA) can be used not only in killed vaccines but also in live vaccines like the Montanide-adjuvanted IB vaccines, providing a higher immune response and protection rate than the unadjuvanted IB vaccines [111].

Oil-in-water (O/W) emulsions, formed by dispersing oil droplets in the aqueous phase, resemble W/O in stimulating innate immunity and adaptive immunity; however, they do not create an antigen depot at the injection site [112]. Experimental evidence indicates that MF59 (a squalene O/W vaccine adjuvant) elicits stronger cellular immunity against the AI virus in mice than aluminum hydroxide and calcium phosphate [113]. However, when tested in a chicken model, the inclusion of MF59 (O/W) adjuvants in AI and ND vaccines showed a lower adjuvanticity effect compared to the nano-aluminum salts [114]. While this study did not specifically compare the MF59 with traditional aluminum salts, this research highlights the potential of nano-alum to serve as a potent adjuvant when used in nanoparticulate form.

Water-in-oil-in-water (W-O-W) emulsion is a unique type of emulsion that is composed mainly of three phases: internal water phase surrounded by a middle oil layer with external water cladding [115]. This particular adjuvant is characterized by its ability to induce long-lasting immunity, along with effective broad immune potentiation. Experimental data indicate that W-O-W induces potent systemic immunity and protection in mice vaccinated by rabies adjuvanted by (W-O-W) compared to vaccines adjuvanted by other types of adjuvants [116]. Overall, although emulsion adjuvants are more potent than mineral salts, they could be carcinogenic to both humans and animals. Consequently, efforts have been made to overcome this side effect by replacing the carcinogenic mineral oils with safer metabolizable ones, which, in fact, resulted in adjuvants that are safe (non-carcinogenic) but less potent in their efficacy [117].

#### 3.1.3. Immune-Stimulating Complexes (ISCOMs)

ISCOMs are novel vaccine adjuvants that are known for their efficient antigenic presentation to the immune system. They are cage-like structures that contain saponins, cholesterol, and phospholipids. The purified forms of ISCOMs have been shown to possess an immunostimulatory potential by triggering Th1, cytotoxic T cell (CTL), and, to some extent, Th2 responses [118]. Experimental studies revealed that ISCOMs can be used as an efficient adjuvant in poultry vaccines such as *M. gallisepticum* [119] and *E. acervulina* [120]. In another study, utilizing Quil-A, a component of saponin, and chitosan (QAC) as an adjuvant for a DNA vaccine expressing IBV nucleocapsid (N) protein enhanced its immunogenicity and decreased viral shedding post-challenge [121]. Nevertheless, it is essential to highlight that ISCOMs have shown toxic effects in both rats and mice, albeit with lower intensity in other animal species [122]. Therefore, a thorough investigation of this side effect is crucial to ensure the safety of using ISCOMs as commercial poultry vaccine adjuvants.

Overall, the numerous limitations associated with conventional adjuvants, including relatively low immunostimulant effects, compromised meat quality due to granuloma formation at the injection site, and potential toxicity, underscore the necessity for exploring alternative adjuvants that are both safe and potent.

### 3.2. Recent Trends in Vaccine Adjuvants

#### 3.2.1. Toll-like Receptor (TLR) Agonists

During the early 1940s, Freund and his colleagues observed that the inclusion of killed *Mycobacteria* in mineral oil emulsions led to increased antibody production in rabbits and guinea pigs [123]. However, at that time, the mechanism underlying this effect remained unknown. With advancements in modern technology, it was revealed that components derived from bacteria, termed pathogen-associated molecular patterns (PAMPs), uniquely stimulate certain receptors expressed by the immune cells, known as PRRs [124,125,126,127]. The interactions between PAMPS and PRRs initiate intracellular signaling pathways leading to cellular proliferation and differentiation and secretion of immunomodulatory substances, such as cytokines, chemokines, and antimicrobial peptides [128,129].

To date, ten TLRs (the most extensively studied family of PRRs) have been discovered in poultry [130]. These TLRs are classified into subfamilies that primarily recognize distinct PAMPs. For instance, TLR4 usually recognizes LPS [131], while unmethylated CpG oligodeoxynucleotide (ODN) is recognized by TLR21 [130]. Until recently, the role of TLRs in mediating immune responses was not fully understood.

Advanced biotechnology and immune engineering aided in unveiling the PAMP-TLR interactions as well as in designing and creating macromolecules or synthetic forms of TLR agonists [132]. Synthetic macromolecules and their assemblies serve as a foundational element in TLR agonist technology. They are more specific in their mechanism of action and easily reproducible, and their higher-order structure allows for nano-scale control [133]. Crucially, numerous studies highlighted that these synthetic ligands have shown significant promise in effectively serving as vaccine adjuvants, exhibiting effects similar to their natural counterparts. For example, the TLR2 is usually activated by the lipoproteins of Gram-positive bacteria, but it can also be activated by Pam3CSK4 [127,130]. Additionally, the CpG DNA is the natural agonist of TLR21, while the CpG DNA assemblies, synthetic single-stranded DNA, can also possess TLR21 enhancement activity [134,135,136].

TLR agonists can be applied via injection or oral/intranasal routes [137], implying that they can be used as mucosal and systemic adjuvants. Initially, TLR agonists were given concurrently with subunit vaccines but in separate formulations. However, advancements in biotechnology strategies have allowed vaccine producers to merge both TLR agonists and subunit proteins into a single fused vaccine for a single administration [138]. This has facilitated the antigen and TLR agonist uptake by APCs and promoted ideal major histocompatibility complex (MHCII)-Th responses [93]. One commonly employed TLR agonist as a vaccine adjuvant is LPS, which constitutes a structural element of the surface membrane of Gram-negative bacteria. LPS is a TLR4 agonist that can enhance APCs to secrete cytokines, including IL-1*β*, IL-12, IL-18, IL-23, and IL-27. These cytokines trigger the activation of Th1 cells, which consequently promote both cellular and humoral immunity against different viruses and other intracellular pathogens [97]. Likewise, CpG ODN interacts with TLR21 and promotes Th1 response by eliciting cytokine secretion, including IL-12 and IFN-γ. Co-administration of CpG ODN with inactivated AIV as an antigen elicited a stronger immunity than that induced by the antigen alone following intranasal application [139,140]. It has also been reported that a fusion of CpG ODN with chicken anemia virus recombinant protein boosted chicken immune responses [141]. Despite the increasing interest in incorporating TLR agonists in veterinary vaccine production, a notable challenge arises due to the potential risk of over-activating the innate immune system through PRRs, which could result in septic shock [142].

#### 3.2.2. Cytokines

Cytokine genes and proteins have been extensively explored as veterinary vaccine adjuvants [143]. Generally, cytokines play a crucial role in orchestrating host immune responses. While some cytokines act as stimulators of specific immune cells, others function as general immune promotors. For example, IL-2, IL-12, and IFN-*γ* induce Th1 cell response, which is responsible for cell-mediated immunity; however, IL-4, IL-5, and IL-10 induce Th2 cell response and antibody production [98]. The use of cytokines as vaccine adjuvants has been explored in numerous research trials with varying degrees of success, including co-administration of IL-18 with ND virus [141], co-expression of both HA of AIV and chicken IL-18 in recombinant fowlpox vaccine [144], and integration of IL-7 in DNA vaccine against IBD [145]. The addition of recombinant rHis-ChIL-18 to *C. perfringens* α-toxoid and ND virus vaccines resulted in significantly higher antibody titers compared to vaccines adjuvanted with aluminum and chitosan [146].

While cytokines offer numerous advantages that position them at the forefront among adjuvants, certain drawbacks have hindered their commercialization. One limitation is their species specificity; specific cytokines may have distinct effects in one species but may not have the same impact in others. For instance, IL-10 can induce both Th1 and Th2 in mice, while no such effect was observed in cattle [144]. Moreover, some cytokines are less stable when mixed with vaccines and are toxic when administered in large doses. Administering cytokines as an adjuvant with vaccines in relatively high doses can result in adverse effects on recipients, such as shock, autoimmunity, immunosuppression, and, in severe cases, death. Conversely, the administration of low doses has been found to be inefficient [99].

#### 3.2.3. Nano-Adjuvants

One drawback in commercializing certain adjuvants, such as TLR agonists and cytokines, is the challenge of effectively delivering them to mucosal surfaces. This limitation has recently been addressed through the use of nano-carriers. These nanoparticles not only facilitate the mucosal delivery and controlled release of adjuvants but also demonstrate the ability to reduce the required effective dosage while enhancing their immunostimulatory effects [147]. It is important to note that the efficacy of these nanoparticles relies on factors such as size and surface charge, with nanoparticles exhibiting a more significant potential to function as adjuvants than microparticles. The diverse types of nano-carriers and their mechanisms of action have been extensively reviewed elsewhere [147].

Studies in chickens have shown that nano-adjuvants hold significant promise for enhancing vaccine immunogenicity. For instance, incorporating aluminum hydroxide into chitosan nanoparticles and administering them with ND- and AIV-inactivated vaccines resulted in enhanced immunogenicity. This was evidenced by elevated levels of antibody titers, serum IgG, IL-4, and IFN-γ compared to the immunogenic response generated by commercial inactivated vaccines when administered alone [148]. Along similar lines, encapsulation of various TLR ligands, including LPS (TLR4 ligand), CpG ODN (TLR21 ligand), and Pam3CSK4 (TLR2 ligand) in PLGA polymeric nanoparticles induced higher and prolonged innate responses both in vivo and in vitro, suggesting their capability as stand-alone prophylactic agents against pathogens [149]. Similar findings were noted when CpG ODN was encapsulated with carbon nanotubes [150]. Aerosol administration of an inactivated AIV vaccine containing PLGA-encapsulated CpG ODN 2007 yielded superior mucosal responses compared to non-encapsulated CpG ODN 2007 [140]. In a vaccination-challenge trial, the administration of PLGA-encapsulated CpG ODN in conjunction with an inactivated AIV resulted in a substantial reduction in virus shedding, surpassing the efficacy of the vaccine alone [151]. In terms of its effectiveness against bacterial pathogens, the oral administration of PLGA-encapsulated CpG ODN 2007 to broiler chickens led to heightened immune responses in the ileum and cecal tonsils, along with a decrease in *Campylobacter* colonization, compared to non-encapsulated CpG ODN 2007 [12,13]. While nanoparticles exhibit considerable promise as vaccine adjuvants, there are some constraints associated with their application in poultry. The pros and cons of nano-adjuvants are depicted in Figure 4.

## 4. Vaccine Stabilization

Vaccines may undergo degradation if exposed to adverse environmental conditions, such as elevated temperatures beyond the recommended range during storage and transportation. Furthermore, the effectiveness of certain vaccines might diminish upon reconstitution and repeated freeze–thaw cycles due to structural alterations. Hence, maintaining the stability of vaccines is essential to safeguard their potency and effectiveness against physical and chemical degradation caused by temperature fluctuations throughout the processes of manufacturing, distribution, storage, and application. The selection of stabilizers in vaccine formulation is primarily determined by the type of vaccine, the presence of an adjuvant, and the expected shelf life. Commonly employed stabilizers consist of either protein components (such as peptides, amino acids, human serum albumin, lactalbumin, gelatin, and polygeline) or sugar and sugar alcohols (like sucrose, trehalose, sorbitol, mannitol, and lactose) [152]. Despite the inclusion of these stabilizers, if a cold-chain system is not provided for storage and transportation, particularly for live attenuated and subunit vaccines, their effectiveness can be compromised. Yet, preserving the cold chain poses significant challenges, particularly in developing countries with a crucial vaccine demand. Given this, the thermal stability of many available vaccines, particularly those designed to combat internationally significant animal diseases, becomes a critical concern.

Indeed, freeze-drying/lyophilization methods stand out as the most prevalent approach for enhancing vaccine stability and have been extensively employed for the production of commercially modified live vaccines, including those for *M. gallisepticum*, ILT, NDV, and IB. While freeze-drying has proven effective in developing thermally stable vaccines, it is crucial to note that not all vaccines are compatible with this process [153]. Additionally, the potency of lyophilized vaccines may experience a decline after reconstitution [153].

Recent explorations into alternative methods for freeze-drying, such as spray-drying/foam drying, have revealed promising alternatives that eliminate the need for freezing or high vacuum. These methods transform liquid vaccines into dispersible particles, facilitating the production of bulk powder vaccines. While spray drying appears promising for vaccine stabilization, it has some drawbacks. Notably, there is an elevated risk of antigen exposure to shear stress during atomization, increased temperatures during drying, and the potential formation of air-water interfaces during droplet formation, which could lead to antigen denaturation [154]. A previous study highlighted the potential of developing a one-step spray-dried dry powder formulation for an attenuated live NDV vaccine, demonstrating the advantageous role of stabilizers like mannitol, trehalose, polyvinylpyrrolidone (PVP), and bovine serum albumin (BSA) in preserving vaccine titers for 10 months at temperatures of 6 and 25 °C [155]. While the deployment of technology, including the provision of specialized freezers, dry ice, and lyophilizers, has addressed some of these challenges, ongoing research is focused on exploring alternative stabilization methods to ensure adequate vaccine potency at ambient temperatures.

Utilizing reverse genetics technology, Tan and colleagues have recently created a recombinant thermostable NDV vector vaccine that expresses multiple epitope cassette S-T/B (rLS-T-HN-T/B) of the IBV [156]. The thermostability of the NDV was achieved by introducing the HN gene from the TS09-C strain into the LaSota strain, resulting in a thermostable avirulent recombinant strain known as rLS-T-HN. This vaccine, with its proven stability in liquid form for 16 days at 25 °C, offers the potential to be administered through drinking water and as a spray, eliminating the requirement for a cold chain during its distribution, storage, and application. While these methods have demonstrated success in experimental settings, they have not yet been applied to any licensed vaccines. Figure 6 presents a schematic diagram depicting various stabilization methods.

## 5. Vaccine Delivery

Vaccines can be administered through several routes, including oral and parenteral routes such as intramuscular, subcutaneous, and intradermal methods [157]. Effective vaccine administration strategies are crucial to maximizing the immunogenic window and lowering vaccination-related hazards [9,158]. The development of optimal vaccine delivery routes helps to achieve high efficacy of the vaccine with minimal side effects and obtain adequate adaptive immune responses [159]. The delivery methods for commercial vaccines are included in Appendix A.

### 5.1. Conventional Vaccine Delivery Methods

The conventional vaccine delivery system in poultry involves routes such as oral, nasal spray, wing web injection, and intramuscular and subcutaneous injections [160,161] (Figure 7). The oral route for vaccination is the most common and non-invasive method that stimulates intestinal mucosal response [162]. Delivering vaccines in the form of oronasal spray or aerosol through the nasal cavity allows for better targeting of the mucosal lining and Harderian gland located in the upper respiratory tract of chickens. This method of administration provides enhanced protection against actual respiratory diseases such as AI and ND [163].

The intramuscular and subcutaneous routes are among the most commonly employed methods for vaccination against bacterial diseases in poultry [10]. The intramuscular administration of lysogenic strain, mainly the aromatic-dependent mutants of *Salmonella gallinarum*, was able to confer significant protection against fowl typhoid. Additionally, a vaccine containing an attenuated mutant of *S*. Typhimurium induced a significant immune response against salmonellosis in poultry when administered intramuscularly [164].

Administration of the vaccine through the wing web stab method is one of the oldest methods of vaccine administration in poultry [165]. This type of vaccine is most commonly used against pathogens like FPV [166]. In this method, a specialized applicator needle with two tongs is dipped in the vaccine and stabbed into the wing web of the birds, ensuring that no blood vessels are punctured [167]. A comparable protection was observed against FPV in chickens vaccinated with a commercial vaccine containing a non-haemagglutinating FPV administered orally and through the wing web stab method [168].

### 5.2. Recent Trends in Vaccine Delivery Methods

Emerging technologies like nanotechnology have gained increased significance in creating vaccines utilizing noncarrier-based delivery systems [169]. The effectiveness of administering the oral chitosan (CNP) vaccine against *Salmonella* was evaluated in broiler chicken; the mass administration of the CNP oral vaccine significantly stimulated mucosal immune response and increased the OMPs-specific IgY [170]. Another study conducted by Zhao et al. demonstrated that the NDV-encapsulated chitosan nanoparticles containing lentogenic virus vaccine (strain LaSota) provided better protection against ND in specific-pathogen-free chicken than the traditional inactivated NDV vaccine [171]. Along similar lines, Hajam et al. suggested that intranasally delivered CS NP vaccines can induce robust immune responses against AIV more efficiently, demonstrated by increased systemic IgG and secretory IgA antibody response along with cross-reactive neutralizing antibodies and T-cell response [172]. The oral administration of polymer nanoparticles (PNP) containing the OMPs and flagellar proteins in layer chicken resulted in significant increases in OMPs-specific IgG response and secretion of Th1 cytokine IFN-γ in the serum. Moreover, there was an enhanced CD8+/CD4+ cell ratio in the spleen, as well as an increase in OMP-specific lymphocyte proliferation [173].

A novel approach in biotechnology is the use of in ovo vaccination, which involves making a small hole in the blunt end of a chicken egg using an obliquely pointed needle and then delivering a small amount of vaccine into the amniotic cavity using a smaller needle [174]. In ovo administration is a way of delivering the vaccine into the egg to benefit the developing late-stage embryos before the completion of the incubation period [175]. It is usually performed at a limited time frame of the incubation when the eggs are being transferred from the setter to the hatcher units, giving advantages such as the capability to vaccinate a larger number of eggs, cost-effective, low human errors, and faster and better immune responses [176]. Generally, the vaccine is injected into the amniotic fluid, which is then ingested by the embryo, allowing for “oral immunization” at an early stage [177].

The efficacy of the vaccine and safety of the embryo depends on the sites of the vaccine administration, including the air cell (AC), allantoic sac (AL), amnion/amniotic fluid (AM), embryonic body (EM), and yolk sac [178] (Figure 8). For example, vaccines administered into the AC and AL generally show low efficacy as compared to those injected into the AM [175]. Experiments conducted by Alqhtani and colleagues have suggested that the embryonic mortality in the case of vaccine delivered into the AC and AL is lower; however, the development of humoral immunity is not sufficient and is significantly lower as compared to those injected into the AM [179]. For example, the efficacy of the MD vaccine delivered into the AC and AL was very low regardless of the frequency of application and type of egg used [180]. Additionally, the vaccines administered through the AM are considered to be more efficacious because of the high systemic assimilation in the embryo as the substances in the amnion are readily absorbed, digested, and distributed throughout the embryonic body [181]. Interestingly, experiments conducted by Patricia and colleagues on the in ovo delivery of MD vaccine confirmed that the injections into the AM yielded more optimum hatch immunity than other routes, such as AC route [180].

Vaccines can be directly injected into the embryonic body, which is further divided into subcutaneous, intramuscular, intracranial, intraorbital, or intra-abdominal regions [182]. The vaccine administration through the intramuscular route provides equal protection as the amniotic route, but the injection through the intracranial, intraorbital, or intra-abdominal regions can cause excessive embryonic trauma and high embryonic mortality [183].

The effectiveness of *Campylobacter jejuni*-modified outer membrane vesicles (OMVs) was assessed in 18-day-old chicken embryos. The study revealed overexpression of wtOMVs and wtCjaA, along with minimal cecal colonization of *C. jejuni* 14 days post-hatching [14,177]. Similarly, in another study, in ovo administration of *Eimeria* profilin and *C. perfringens* NetB protein with IMS adjuvants demonstrated increased body mass gain and decreased level of proinflammatory cytokines and chemokines with decreased disease pathology [184]. The in ovo administration of silver nanoparticles at embryonic day 18 did not affect hatchability and enhanced chicks’ resistance to NDV [185]. Another study evaluated the impact of cathelicidin (D-CATH-2) on avian pathogenic *Escherichia coli* when administered through in ovo injection on the 18th day of embryonic life and via intramuscular injection on the first- and fourth-days post-hatch, and the mortality of the chicken was measured after the 7th day post-hatch [186]. The results showed a 30% decrease in mortality, a 63% decrease in morbidity, and a more than 90% decrease in bacterial load in the respiratory system [186]. The in ovo delivery of bioactive substances, including vaccines, has been reviewed elsewhere [187].

In sum, achieving successful industrial translation of a new generation of vaccines necessitates ongoing research and the incorporation of scientific progress in formulating vaccines. This includes the careful selection of antigens, adjuvants, and stabilizers, along with optimizing production processes for scalability and cost-effectiveness. The design of practical and convenient administration methods is equally essential. Moreover, these vaccines must be developed in compliance with international regulatory standards to access the market and garner acceptance from poultry producers.

## 6. Conclusions and Future Prospects

Technological advancements over the past few decades have played a pivotal role in transforming the creation of novel vaccines and utilizing nanoparticles as delivery systems. The identification of highly conserved antigenic targets, facilitated by bioinformatics approaches, coupled with the success in developing multi-epitope vaccines that encompass various antigenic targets, holds the potential for broader protection against emerging diseases and expedites development timelines. For instance, a contemporary genome-based strategy, known as reverse vaccinology, has exhibited promise by identifying potential vaccine candidates (relevant protein antigens) through computational analyses of a pathogen’s proteome. Additionally, integrating bioinformatics with immunogenetics has revolutionized vaccine design, contributing to improvements in effectiveness, safety, specificity, and thermodynamic stability compared to conventional vaccine development approaches. Despite its success in developing effective vaccines against bacterial pathogens in humans, such as *Neisseria meningitidis*, reverse vaccinology is still under investigation for poultry vaccine development. Nevertheless, ongoing research is exploring its application in the development of multi-epitope vaccines targeting bacterial pathogens such as *M. gallisepticum*, *C. jejuni*, and *C. perfringens*, viral pathogens such as IBD and chicken anemia virus, and parasitic pathogens such as *Eimeria* species in poultry.

While not yet implemented in poultry, the success of the nanoparticle delivery system demonstrated in humans holds promise for the potential application of this technology in mucosal vaccine delivery for animals, including chickens. Ongoing research is actively exploring the feasibility of this approach for delivering vaccines to chickens, taking into consideration the relatively high costs associated with particle production. Lastly, the persistent challenge faced by the poultry industry stems from the continual emergence and re-emergence of infectious diseases, particularly in the “post-antibiotic era”, necessitating a continuous improvement of vaccine effectiveness, stability, and delivery.

## Figures and Tables

**Figure 1 vaccines-12-00134-f001:**
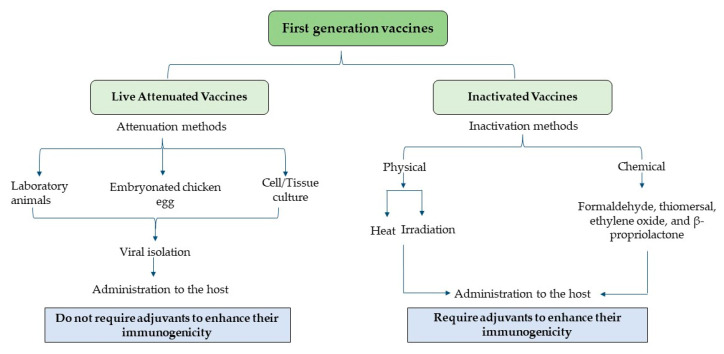
First-generation vaccines.

**Figure 2 vaccines-12-00134-f002:**
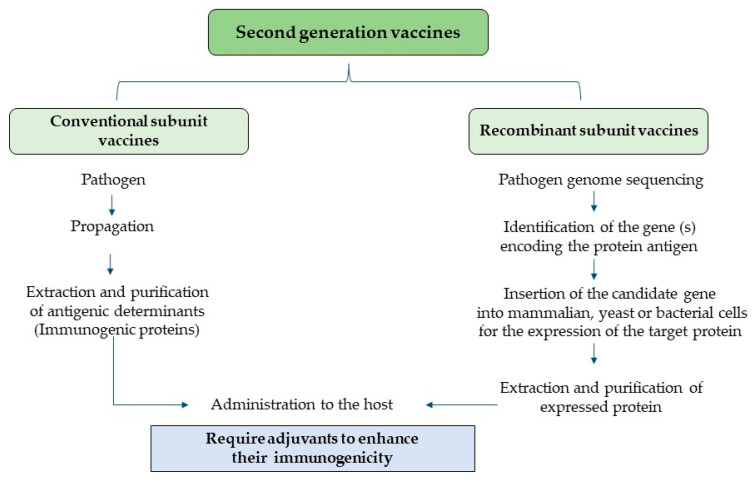
Second-generation vaccines.

**Figure 3 vaccines-12-00134-f003:**
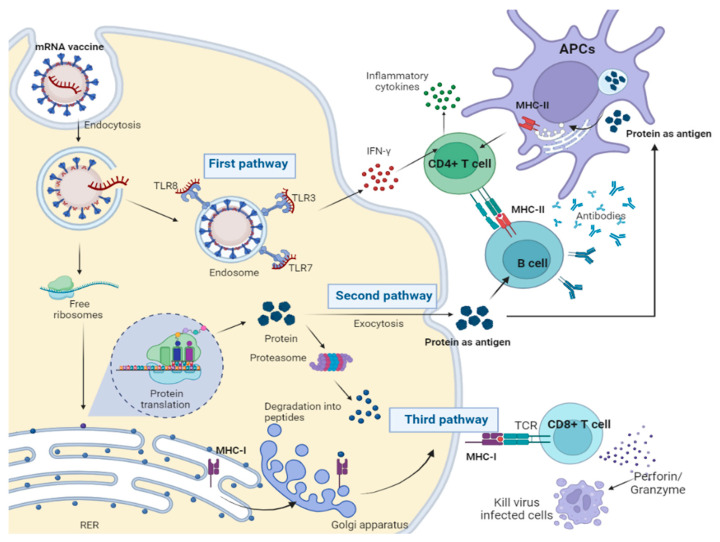
Mechanisms of action of mRNA vaccines.

**Figure 4 vaccines-12-00134-f004:**
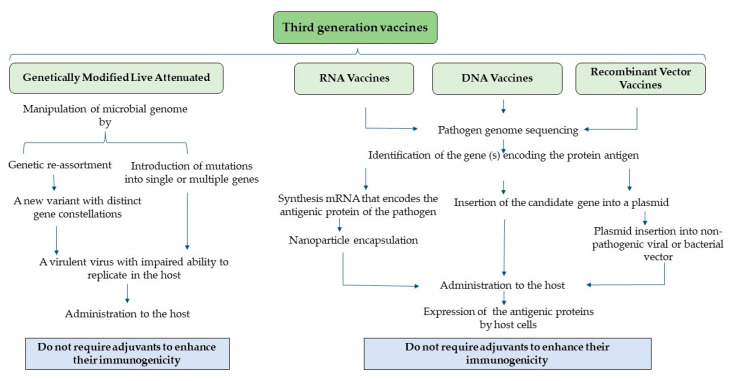
Third-generation vaccines.

**Figure 5 vaccines-12-00134-f005:**
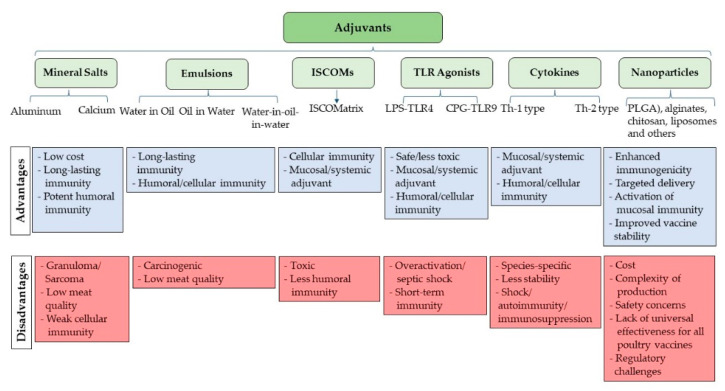
Different categories of adjuvants.

**Figure 6 vaccines-12-00134-f006:**
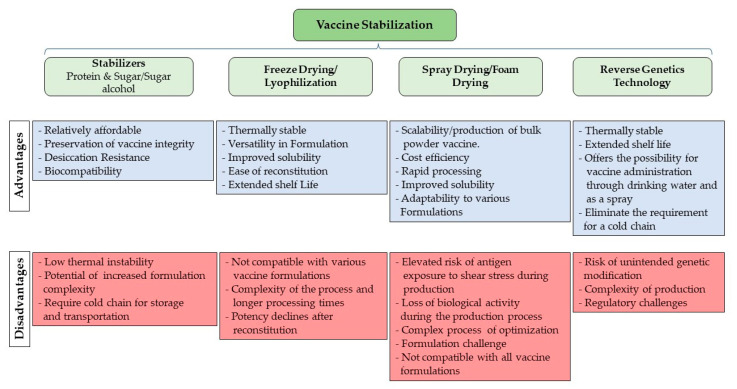
Vaccine stabilization methods.

**Figure 7 vaccines-12-00134-f007:**
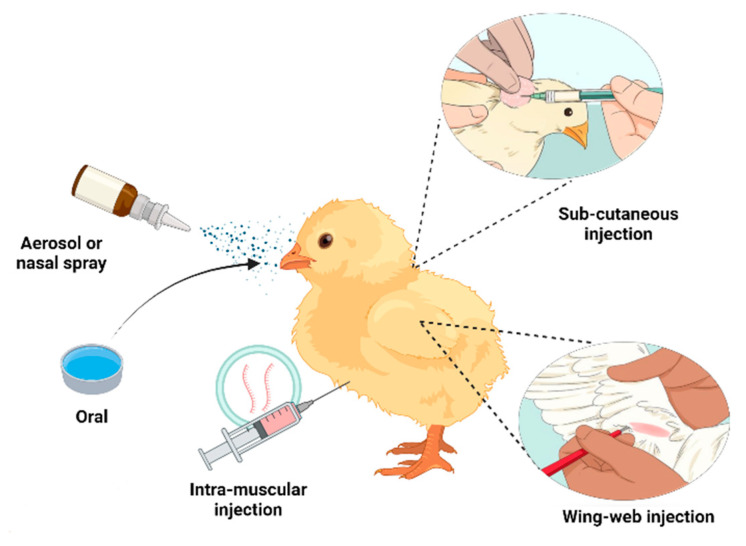
Vaccine administration routes.

**Figure 8 vaccines-12-00134-f008:**
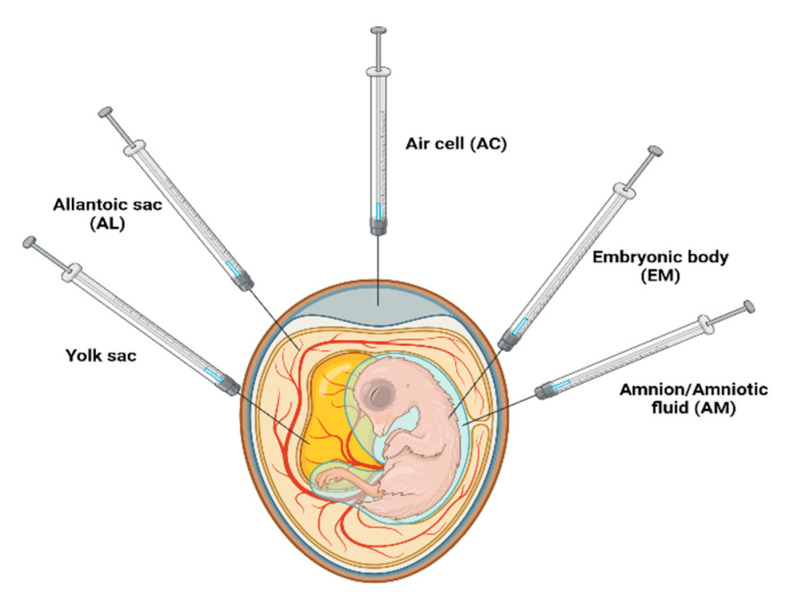
Sites for in ovo administration of vaccines.

**Table 1 vaccines-12-00134-t001:** A list of avian species commonly involved in the poultry industry.

Species (English Name)	Latin Name
Chicken	Gallus gallus domesticus
Duck	Anas platyrhynchos domesticus
Turkey	Meleagris gallopavo
Goose	Anser domesticus
Quail	Coturnix coturnix
Pigeon	Columba livia domestica
Guinea Fowl	Numida meleagris
Emu	Dromaius novaehollandiae
Pheasant	Phasianus colchicus
Ostrich	Struthio camelus

## Data Availability

Not applicable.

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
