# Peer review of "Advances in Poultry Vaccines: Leveraging Biotechnology for Improving Vaccine Development, Stability, and Delivery"

_vaccines, 2024, doi:10.3390/vaccines12020134_

Round 1

Reviewer 1 Report

Comments and Suggestions for Authors

Khaled Abdelaziz et al. submitted an interesting review about biotechnology application in poultry vaccines. The review subject was interesting, and the topic might arouse a certain impact in its field. However, there were several flaws deterring the acceptance of this paper. Hence, a Major Revision must be performed before a second review. Detail comments:

1.       As the audience for Vaccines might not be familiar with the area of poultry, please provide a list of animal species (English and Latin names) involved in poultry industry.

2.       Line 90-91: Too simple. As the last part in Introduction, the whole picture of this review should be well demonstrated.

3.       An important point: In Section 2, a few generations of vaccines had been discussed. The reviewer suggested to consider that nanotechnology-assisted vaccines might also be a new generation of vaccine. Please add relevant information or make proper explanations.

4.       Following question 3, as well in Section 3.2, the nano-adjuvants might be trending nowadays. Please add relevant information or make proper explanations.

5.       In Section 4, before introducing the stabilization methods, it was advisable to add a paragraph to explain why vaccines were unstable. Moreover, the stabilization methods could be summarized in a tabular manner.

6.       A schematic illustration was recommended to supplement in Section 5.2, in parallel to Figure 5 in Section 5.1.

7.       Before the Conclusion Section, a Discussion Section regarding the industrial translation aspects could be added.

8.       Please double-check the format of References.

Author Response

Khaled Abdelaziz et al. submitted an interesting review about biotechnology application in poultry vaccines. The review subject was interesting, and the topic might arouse a certain impact in its field. However, there were several flaws deterring the acceptance of this paper. Hence, a Major Revision must be performed before a second review. Detail comments:

  1. As the audience for Vaccinesmight not be familiar with the area of poultry, please provide a list of animal species (English and Latin names) involved in poultry industry.

We have included a table in the introduction section providing a list of various avian species involved in the poultry industry (please see line 39-42.

Table 1 provides a list of various animal species commonly involved the poultry industry.

Species (English name)

Latin name

Chicken

Gallus gallus domesticus

Duck

Anas platyrhynchos domesticus

Turkey

Meleagris gallopavo

Goose

Anser domesticus

Quail

Coturnix coturnix

Pigeon

Columba livia domestica

Guinea Fowl

Numida meleagris

Emu

Dromaius novaehollandiae

Pheasant

Phasianus colchicus

Ostrich

Struthio camelus

  1. Line 90-91: Too simple. As the last part in Introduction, the whole picture of this review should be well demonstrated.

Our statement has been modified to the following “This review presents various novel approaches that have incorporated the use of new technologies for improving vaccine immunogenicity and efficacy, as well as the creation of potent adjuvants. Furthermore, the role of biotechnology in employing advanced and scalable methods for vaccine stabilization and delivery will be discussed”.

  1. An important point: In Section 2, a few generations of vaccines had been discussed. The reviewer suggested to consider that nanotechnology-assisted vaccines might also be a new generation of vaccine. Please add relevant information or make proper explanations.

We have included the following section “ Due to the delicate nature of the RNA, delivery systems have been devised to safeguard and stabilize the mRNA constructs from degradation during applications and enhancing its bioavailability, thereby resulting in a robust immune response. Recent breakthroughs in cutting-edge nanotechnology have revolutionized vaccine delivery platforms. For instance, nanoparticle (NP)-based technologies have emerged as promising replacements for older vaccine delivery methods. Substantial progress has been achieved in the development and application of delivery technologies, including lipid-based nanoparticles (LNPs), carbon nanotubes, polyplexes, polymeric nanoparticles, hydrogel beads, and colloidal nanoparticles made from Generally Recognized As Safe (GRAS) polysaccharides and proteins (e.g., alginate, chitosan, and gelatin) and other delivery methods, such as squalene-based cationic nano-emulsions [77,81].

We have also updated this section with recently published data (lines 420-431; highlighted in yellow).

  1. Following question 3, as well in Section 3.2, the nano-adjuvants might be trending nowadays. Please add relevant information or make proper explanations.

We have added the below section under the adjuvants and updated the schematic diagram.

3.2.3 Nano-adjuvants

One drawback in commercializing certain adjuvants, such as TLR ligands and cytokines, is the challenge associated with effectively delivering them to mucosal surfaces. This limitation has recently been addressed through the use of nano-carriers. These nanoparticles not only facilitate the mucosal delivery and controlled release of adjuvants but also demonstrate the ability to reduce the required effective dosage while enhancing their immunostimulatory effects [147]. It is important to note that the efficacy of these nanoparticles relies on factors such as size and surface charge, with nanoparticles exhibiting a more significant potential to function as adjuvants than microparticles. The diverse types of nano-carriers and their mechanisms of action have been extensively reviewed elsewhere [147].

Studies in chickens have shown that nano adjuvants hold significant promise for enhancing vaccine immunogenicity. For instance, incorporating aluminum hydroxide into chitosan nanoparticles and administering them with ND- and AIV-inactivated vaccines resulted in enhanced immunogenicity. This was evidenced by elevated levels of antibody titers, serum IgG, interleukin-4 (IL-4), and interferon-gamma (IFN-γ) compared to the immunogenic response generated by commercial inactivated vaccines when administered alone. [148]. Along similar lines, encapsulation of various TLR ligands, including LPS (TLR4 ligand), CpG ODN (TLR21 ligand), and Pam3CSK4 (TLR2 ligand) in PLGA polymeric nanoparticles induced higher and prolonged innate responses both in vivo and in vitro, suggesting their capability as stand-alone prophylactic agents against pathogens (Alkie et al., 2017). Similar findings were noted when CpG ODN was encapsulated with carbon nanotubes [149]. Aerosol administration of an inactivated AIV vaccine containing PLGA-encapsulated CpG ODN 2007 yielded superior mucosal responses compared to non-encapsulated CpG ODN 2007 [140]. In a vaccination-challenge trial, the administration of PLGA-encapsulated CpG ODN in conjunction with an inactivated AIV resulted in a substantial reduction in virus shedding, surpassing the efficacy of the vaccine alone [150]. In terms of its effectiveness against bacterial pathogens, the oral administration of PLGA-encapsulated CpG ODN 2007 to broiler chickens led to heightened immune responses in the ileum and cecal tonsils, along with a decrease in Campylobacter colonization, compared to non-encapsulated CpG ODN 2007 [12,13]. While nanoparticles exhibit considerable promise as vaccine adjuvants, there are some constraints associated with their application in poultry.  The pros and cons of nano adjuvants are depicted in Figure 4.

  1. In Section 4, before introducing the stabilization methods, it was advisable to add a paragraph to explain why vaccines were unstable. Moreover, the stabilization methods could be summarized in a tabular manner.

Thank you for the suggestion. We have included the following paragraph in Section 4: “Vaccines may undergo degradation if exposed to adverse environmental conditions, such as elevated temperatures beyond the recommended range during storage and transportation. Furthermore, the effectiveness of certain vaccines might diminish upon reconstitution and repeated freeze-thaw cycles due to structural alterations. Hence, maintaining the stability of vaccines is essential to safeguard their potency and effectiveness against physical and chemical degradation caused by temperature fluctuations throughout the processes of manufacturing, distribution, storage, and application.”

We have also included a schematic diagram depicting various methods of vaccine stabilization (please see Figure 6).

  1. A schematic illustration was recommended to supplement in Section 5.2, in parallel to Figure 5 in Section 5.1.

We have included an illustration showing different sites for in ovo administration of vaccines (please see figure 8).

  1. Before the Conclusion Section, a Discussion Section regarding the industrial translation aspects could be added.

We have added the following paragraph before the conclusion “In sum, achieving successful industrial translation of new generation of vaccines necessitates ongoing research and the incorporation of scientific progress in formulating vaccines. This includes the careful selection of antigens, adjuvants, and stabilizers, along with optimizing production processes for scalability and cost-effectiveness. The design of practical and convenient administration methods is equally essential. Moreover, these vaccines must be developed in compliance with international regulatory standards to access the market and garner acceptance from poultry producers.”

  1. Please double-check the format of References.

Thank you. We have fixed the format of references.

Reviewer 2 Report

Comments and Suggestions for Authors

The review is interesting and described in a logical and sequential manner. The only thing that is recommended to the authors is that they include some figures related mainly to the way in which third-generation vaccines carry out their mechanism of action, specifically all those that use nanoparticulate systems as carriers.

Some suggestions are shown below:

L.257. "Viral vector vaccines"

L.257. "superior safety profiles due to a gen ..."

L.440. "Why were biopolymers such as chitosan not included in the category? or are those not important?"

Author Response

The review is interesting and described in a logical and sequential manner. The only thing that is recommended to the authors is that they include some figures related mainly to the way in which third-generation vaccines carry out their mechanism of action, specifically all those that use nanoparticulate systems as carriers.

Thank you for the suggestion. We have included a figure illustrating the mechanisms of action of mRNA vaccines.

Some suggestions are shown below:

L.257. "Viral vector vaccines"

We have made the requested corrections.

L.257. "superior safety profiles due to a gen ..."

We have made the requested corrections and rephrased the whole statement.

L.440. "Why were biopolymers such as chitosan not included in the category? or are those not important?"

We have added the below section under the adjuvants. The use of chitosan as a nano-carrier has also been highlighted across the manuscript (Lines 84, 382, 552, 623, 647, 775, 778)

3.2.3. Nano-adjuvants

One drawback in commercializing certain adjuvants, such as TLR ligands and cytokines, is the challenge associated with effectively delivering them to mucosal surfaces. This limitation has recently been addressed through the use of nano-carriers. These nanoparticles not only facilitate the mucosal delivery and controlled release of adjuvants but also demonstrate the ability to reduce the required effective dosage while enhancing their immunostimulatory effects [147]. It is important to note that the efficacy of these nanoparticles relies on factors such as size and surface charge, with nanoparticles exhibiting a more significant potential to function as adjuvants than microparticles. The diverse types of nano-carriers and their mechanisms of action have been extensively reviewed elsewhere [147].

Studies in chickens have shown that nano adjuvants hold significant promise for enhancing vaccine immunogenicity. For instance, incorporating aluminum hydroxide into chitosan nanoparticles and administering them with ND- and AIV-inactivated vaccines resulted in enhanced immunogenicity. This was evidenced by elevated levels of antibody titers, serum IgG, interleukin-4 (IL-4), and interferon-gamma (IFN-γ) compared to the immunogenic response generated by commercial inactivated vaccines when administered alone. [148]. Along similar lines, encapsulation of various TLR ligands, including LPS (TLR4 ligand), CpG ODN (TLR21 ligand), and Pam3CSK4 (TLR2 ligand) in PLGA polymeric nanoparticles induced higher and prolonged innate responses both in vivo and in vitro, suggesting their capability as stand-alone prophylactic agents against pathogens (Alkie et al., 2017). Similar findings were noted when CpG ODN was encapsulated with carbon nanotubes [149]. Aerosol administration of an inactivated AIV vaccine containing PLGA-encapsulated CpG ODN 2007 yielded superior mucosal responses compared to non-encapsulated CpG ODN 2007 [140]. In a vaccination-challenge trial, the administration of PLGA-encapsulated CpG ODN in conjunction with an inactivated AIV resulted in a substantial reduction in virus shedding, surpassing the efficacy of the vaccine alone [150]. In terms of its effectiveness against bacterial pathogens, the oral administration of PLGA-encapsulated CpG ODN 2007 to broiler chickens led to heightened immune responses in the ileum and cecal tonsils, along with a decrease in Campylobacter colonization, compared to non-encapsulated CpG ODN 2007 [12,13]. While nanoparticles exhibit considerable promise as vaccine adjuvants, there are some constraints associated with their application in poultry.  The pros and cons of nano adjuvants are depicted in Figure 5.

Reviewer 3 Report

Comments and Suggestions for Authors

Comments on the document

Line 35

The term in ovo should be italicised. This also applies to lines 691, 694, 715, 726, 729 and 736.

Figure 1

Many of these viruses have been passaged through cell culture rather than tissue culture. Very few avian viruses are grown in tissue culture. The exception is probably infectious bronchitis that is propagated in tissue culture of tracheal rings.

The word “virus” is a noun and “viral” is the adjective. Therefore the term is viral isolation.

Line 175

Should probably read and parasitic pathogens by cloning the gene

Line 337

The term is viral nucleoprotein

Line 398

As the abbreviation NE is only used once it may be best if this is in full not abbreviated.

Line 427

The adjuvants do not demonstrate the ability to provoke protective immunity. Studies are carried out in these studies indicate that protective immunity has been provoked. The adjuvants themselves do not participate in demonstration.

Likewise line 430 natural substances do not demonstrate.

Line 436

Emulsions do not exhibit potency. This situation is observed during studies. The adjuvants certainly do not exhibit.

Line 485 again the vaccines did not exhibit.

Line 540 the synthetic leg hands did not demonstrate.

Line 681 again the vaccines did not demonstrate protection. This protection was observed following a series of studies.

Line 720

Should read the vaccine administration through the intramuscular route provides

Line 730

When administered through in ovo injection

Lines 778 to 780 the comment appears to be duplicated.

Comments on the Quality of English Language

Comments on the document

Line 35

The term in ovo should be italicised. This also applies to lines 691, 694, 715, 726, 729 and 736.

Figure 1

Many of these viruses have been passaged through cell culture rather than tissue culture. Very few avian viruses are grown in tissue culture. The exception is probably infectious bronchitis that is propagated in tissue culture of tracheal rings.

The word “virus” is a noun and “viral” is the adjective. Therefore the term is viral isolation.

Line 175

Should probably read and parasitic pathogens by cloning the gene

Line 337

The term is viral nucleoprotein

Line 398

As the abbreviation NE is only used once it may be best if this is in full not abbreviated.

Line 427

The adjuvants do not demonstrate the ability to provoke protective immunity. Studies are carried out in these studies indicate that protective immunity has been provoked. The adjuvants themselves do not participate in demonstration.

Likewise line 430 natural substances do not demonstrate.

Line 436

Emulsions do not exhibit potency. This situation is observed during studies. The adjuvants certainly do not exhibit.

Line 485 again the vaccines did not exhibit.

Line 540 the synthetic leg hands did not demonstrate.

Line 681 again the vaccines did not demonstrate protection. This protection was observed following a series of studies.

Line 720

Should read the vaccine administration through the intramuscular route provides

Line 730

When administered through in ovo injection

Lines 778 to 780 the comment appears to be duplicated.

Author Response

 Line 35

 The term in ovo should be italicised. This also applies to lines 691, 694, 715, 726, 729 and 736.

The term in ovo has been italicized in the indicated lines.

 Figure 1

 Many of these viruses have been passaged through cell culture rather than tissue culture. Very few avian viruses are grown in tissue culture. The exception is probably infectious bronchitis that is propagated in tissue culture of tracheal rings.

Thank you for the thorough review. We have corrected it in the figure to “cell/tissue culture”

 The word “virus” is a noun and “viral” is the adjective. Therefore the term is viral isolation.

We have replaced the word “virus” with “viral”

 Line 175

 Should probably read and parasitic pathogens by cloning the gene

We have added the word “pathogens” after parasitic.

 Line 337

 The term is viral nucleoprotein

Thank you. We replaced the word “virus” with “viral”

 Line 398

 As the abbreviation NE is only used once it may be best if this is in full not abbreviated.

We have replaced the NE abbreviation with the full name “necrotic enteritis.”

 Line 427

 The adjuvants do not demonstrate the ability to provoke protective immunity. Studies are carried out in these studies indicate that protective immunity has been provoked. The adjuvants themselves do not participate in demonstration.

We corrected the statement to the following “Additionally, numerous studies have indicated that adjuvants enhance the immunoactivity of the vaccines”. Please advise if any further changes are required.

 Likewise line 430 natural substances do not demonstrate.

We have revised the sentence to the following “Since the discovery of adjuvants in 1920, numerous materials, spanning from organic and inorganic to synthetic and natural substances, have been thoroughly researched and shown the capacity to act as potent adjuvants”. Please advise if any further changes are required.

Line 436

 Emulsions do not exhibit potency. This situation is observed during studies. The adjuvants certainly do not exhibit.

We have modified the statement in line 463 to the following “emulsion adjuvants exhibit greater potency than aluminum salts, as they can improve vaccine-induced immunity (cellular and humoral)”. Please advise if any further changes are required.

Line 485 again the vaccines did not exhibit.

We have replaced ‘exhibit” with “showed”. Please advise if any further changes are required.

Line 540 the synthetic leg hands did not demonstrate.

We have revised the statement as per your suggestion to the following “Crucially, numerous studies highlighted that these synthetic ligands have shown significant promise in effectively serving as vaccine adjuvants”. Please advise if any further changes are required.

 Line 681 again the vaccines did not demonstrate protection. This protection was observed following a series of studies.

We have revised the statement to the following “Another study conducted by Zhao et al. demonstrated that the NDV encapsulated chitosan nanoparticles containing lentogenic virus vaccine (strain LaSota) provided better protection against ND in specific-pathogen-free chicken than the traditional inactivated NDV vaccine”

 Line 720

 Should read the vaccine administration through the intramuscular route provides.

Thank you. We add the word “route” after intramuscular.

 Line 730

 When administered through in ovo injection

Thank you. The suggested edit has been made.

 Lines 778 to 780 the comment appears to be duplicated.

Thank you. We have revised our statement and removed duplicate comments.

Round 2

Reviewer 1 Report

Comments and Suggestions for Authors

The manuscript had been well improved.